# Versatile Medium Entropy Ti-Based Bulk Metallic Glass Composites

**DOI:** 10.3390/ma15207304

**Published:** 2022-10-19

**Authors:** Tianzeng Liu, Yanchun Zhao, Li Feng, Pan Gong

**Affiliations:** 1State Key Laboratory of Advanced Processing and Recycling of Non-Ferrous Metals, Lanzhou University of Technology, Lanzhou 730050, China; 2Iron and Steel Research Institute, Jiuquan Iron and Steel Corporation, Jiayuguan 735100, China; 3State Key Laboratory of Materials Processing and Die & Mould Technology, School of Materials Science and Engineering, Huazhong University of Science and Technology, Wuhan 430074, China

**Keywords:** bulk metallic glass, phase transformation, mechanical behavior, corrosion wear resistance

## Abstract

An ultra-strong Ti-based bulk metallic glass composite was developed via the transformation-induced plasticity (TRIP) effect to enhance both the ductility and work-hardening capability of the amorphous matrix. The functionally graded composites with a continuous gradient microstructure were obtained. It was found that the austenitic center possesses good plasticity and toughness. Furthermore, the amorphous surface exhibited high strength and hardness, as well as excellent wear corrosion resistance. Compared with the Ti-6Al-4V alloy, bulk metallic glass composites (BMGCs) exhibit better spontaneous passivation behavior during the potential dynamic polarization. No crystallization was observed on the friction surface, indicating their good friction-reduction and anti-wear properties.

## 1. Introduction

Because of the excellent integrated properties, bulk metallic glasses (BMGs) have become a shining rookie of the material world. They can be used as the functional structural materials with excellent physical, chemical and mechanical properties [1,2]. Ti-based BMGs have drawn special interest due to their high specific strength, good corrosive wear resistance and excellent biocompatibility, and are widely used in medicine, environmental protection, energy conservation, aerospace and other military industries [3]. As known, among the traditional biomedical materials, titanium-based alloys have become the first choice of orthopedic and dental implant products due to their good biocompatibility and mechanical properties, high corrosion resistance and relatively low Young’s modulus [4,5]. Biomedical titanium based amorphous alloys possess higher strength and hardness, which allows them to exhibit better wear resistance under dry and wet friction [6].

However, a significant drawback of BMGs includes the limited strain softening with loading, which restricts their application as an advanced structural material. Furthermore, BMGs are significantly less ductile as compared to crystalline alloys. The plastic deformation of crystalline materials typically occurs by dislocation motion. Due to the periodic arrangement of atoms and the long-range translational symmetry, the propagation of dislocations can be carried out at lower energy or stress states. Although dislocations do not exist in BMGs, the plastic deformation of BMGs occur via the rearrangement of localized atoms. This rearrangement requires higher energy or stress than dislocations and is not easy to slip, thus forming localized softening shear bands. The shear band is the main carrier of the plastic deformation of BMGs and typically forms during the percolation of the stress-induced shear transition region (the active atomic cluster motion region) in the nanoscale layer.

The appearance of in-situ BMGCs is predestined. As is well known, transformation-induced Plasticity (TRIP) is an effective way to enhance the work-hardening ability of conventional materials. For instance, TRIP has been widely used in steel and ceramics to enhance their ductility, toughness and work-hardening behavior. Recently, a great amount of remarkable works utilized the TRIP method to develop some novel advanced structural alloys. For instance, this method was used to improve both the strength and ductility in BMGs and high entropy alloys (HEAs) [7,8,9]. Furthermore, some alloy systems have the possibility of producing in situ composites which consist of shape memory crystals and glassy matrix [10,11].

To beat the strength and ductility trade-off, in-situ BMGCs intervene in the proliferation and expansion of the multiple main plastic deformation carrier, shear bands (SBs). However, SBs form during percolation of the localized stress-induced shear transition region in the nanoscale layer and thus usually induce the strain softening as loading. To improve the stain hardening, TRIP-BMGCs introduce the crystals which can realize stress-induced martensite phase transition and twinning to guarantee a strong work-hardening rate and toughening degree. Based on current understanding, for crystalline alloys the increasing stress-induced martensite transformation via dislocations pile-up in the unstable austenite phase, which delays the plastic deformation period and thereby enhances both the strength and plasticity. However, the strong work hardening behavior mechanism, how the amorphous matrix interacts with the crystals, is not fully explained. In this paper, a novel system of the low elastic modulus, ultra-strong Ti-based BMGCs were developed, through TRIP effect of the B2/B19′ dual shape-memory crystals (DSMCs) to enhance both the ductility and work-hardening capability. The functionally graded composites with a continuous gradient microstructure were obtained during the casting solidification. The amorphous surface of the BMGC was found to exhibit high strength and hardness, as well as excellent wear resistance and corrosion resistance. Importantly, since the Ti-based BMGCs exhibit elastic modulus values that are comparable to those for biological organisms, they have potential for use as implant materials. Meanwhile, the austenitic center of the Ti BMGC possesses good plasticity and toughness. Thus, the DSMC toughened Ti-based BMGCs are expected to have a wide range of application as functional and structural materials [12,13,14].

For this work, to better understand the strong work hardening behavior mechanism, as well as the interaction between amorphous matrix and the crystals, a systematic investigation was conducted to fundamentally understand the effect of DSMC on the macro and micro-mechanical behavior, work-hardening mechanism and functional properties of Ti BMGCs. To accomplish this endeavor, the microstructure, compressive properties and micro-mechanical, nano fretting and creep behaviors at the ambient temperature were examined. Thus, the connection between the macro-mechanical behavior and the microstructure of the alloy, the nanofretting behavior and creep strain in different regions of the sample, which are rarely reported, will be studied. Additionally, the work-hardening response and wear corrosion behavior in different mediums were analyzed, and will expand the application of Ti-based BMGCs as functional and structural materials

## 2. Materials and Methods

The ingots of the Ti_40_Ni_40_Cu_20_ (atomic percent) alloys were prepared by mixing the constituent elements with purities >99.99 (weight percent) in a high-frequency induction vacuum furnace with a water-cooled Cu crucible under an argon atmosphere. Each ingot was re-melted at least six times to ensure a homogeneous microstructure. Ti-based cylinder samples with *Φ*3 mm were prepared by suction casting into copper mold. The microstructure of the alloys was studied, using the X-ray diffractometer (XRD, D/max-2400, Rigaku, Tokyo, Japan) with Cu-Kα radiation (40kV-30mA), thermal field-emission scanning electron microscopy (SEM, JSM-5600LV, JEOL, Akishima, Japan) and transmission electron microscopy (TEM, JEM-2010, JEOL, Akishima, Japan) coupled with the selected area diffraction pattern (SADP). The thermal response of the alloy was investigated using a differential scanning calorimeter (DSC, Nietzsche STA449F3, Munich, Germany) under flowing purified argon at a heating rate of 20 K/min.

For compression tests, *Φ*3 mm cylinder samples with a height-diameter ratio of 2:1 were fabricated. The two compression faces of each sample were then carefully polished such that they were parallel to one other. For the mechanical testing experiments, an Instron 3382 electronic-testing machine was employed. Here, samples were subjected to a strain rate of 5 × 10^−4^ s^−1^. The nanoindentation measurements were carried out at room temperature using a Hysitron TI-950 in-situ nanomechanical testing system equipped with a diamond Berkovich tip (Minneapolis, MN, USA). The load resolution and background noise were less than 1 and 30 nN, respectively. Furthermore, the displacement resolution and background noise were less than 0.02 and 0.2 nm, respectively. The specimens were mechanically polished to a mirror finish before nanoindentations. The displacement–load curves of the edge, transition and center zones were obtained by loading and unloading at a constant loading rate of 0.2 mN/s, loading up to the maximum 10 mN for 10 s, then unloading to 0 mN at the same loading rate.

Electrochemical measurements were conducted using the Potentiostat Workstation μAutolab Type III. Tests were carried out using a three-electrode cell equipped with a platinum counter electrode (CE) and an Ag/AgCl reference electrode (RE). The samples were immersed in solutions, where they were used as working electrodes (WE). The tests were performed in 310 K Phosphate-buffered Saline (PBS) solution (8 g/L NaCl, 0.2 g/L KCl, 0.14 g/L NaH_2_PO_4_, 0.2 g/L KH_2_PO_4_, prepared from analytic reagents and distilled water, and diluted with 1mol/L NaOH up to PH 7.3), and 298 K Artificial Sea Water (ASW) (24.53 g/L NaCl, 5.20 g/L MgCl_2_, 4.09 g/L Na_2_SO_4_, 1.16 g/L CaCl_2_, 0.695 g/L KCl, 0.201 g/L NaHCO_3_, 0.101 g/L KBr, 0.027 g/L H_3_BO_3_, 0.025 g/L SrCl_2_, 0.003 g/L NaF, prepared from analytic reagents and distilled water, and diluted with 1mol/L NaOH up to PH 8.2), respectively. The potentiodynamic polarization curves were recorded using a potential sweep rate of 5 mV/s in which the potential ranged from −0.8 to +0.8 V/E. After immersing the specimens for 30 min, the experiments were performed in simulated seawater and simulated body solutions that was exposed to air. Furthermore, the experiment was run until the open-circuit potential (OCP) reached steady-state conditions. All electrochemical measurements were repeated at least three times to verify the reproducibility of the results. The surface linear reciprocating sliding friction and wear tests were carried out on an MFT-R4000 high-speed friction and wear testing apparatus using GCr15 balls with a diameter of 9.6 mm under a normal load of 20N. From the results, the morphology and corrosion products of wear corrosion were determined via a SEM JSM-5600LV with energy dispersive spectrometry (EDS) mapping.

## 3. Results and Discussion

### 3.1. Microstructure

The X-ray diffraction patterns of the as-cast and fracture sample after compressive loading are shown in Figure 1a. As can be seen, the XRD pattern of the as-cast alloy exhibits a diffuse hump for angles ranging from 2*θ* = 35° to 2*θ* = 50°. Moreover, sharp crystal diffraction peaks which represent the B2-Ti (Ni, Cu) and B19′-Ti (Ni, Cu) phases are superimposed over the diffuse scattering peak, indicating that the structure contains both a metallic glass matrix and crystal composite structures. The radius of the Ti, Ni and Cu atoms were also determined to be 0.086, 0.069 and 0.073 nm, respectively. As Cu is larger than Ni, Cu atoms will increase the interatomic spacing when they replace Ni atoms in the B2-TiNi and B19′-TiNi structures, which leads to the formation of the Ti (Ni, Cu) solid solutions. Consequently, the XRD diffraction peaks will shift to lower angles. The TEM selected area electron diffraction (SAED) patterns and HRTEM image of the as-cast Ti_40_Ni_40_Cu_20_ are shown in Figure 1b,c. As mentioned in the X-ray diffraction patterns, the corresponding dispersion halo ring and the diffraction patterns confirm the amorphous and DSMC mixed structure. The high-resolution transmission electron microscopy (HRTEM) and the fast Fourier transform (FFT), further certified the microstructure of DSMC BMGCs. The mixing enthalpies, Δ*H_mix_* Ti-Ni = −35 KJ/mol, Δ*H_mix_* Ti-Cu = −9 KJ/mol and Δ*H_mix_* Ni-Cu = 4 KJ/mol, indicate the stronger bonding between Ti and Ni. Additionally, as the cooling rates significantly increase from inner region of the material to the surface, the thermally-induced B19′-Ti (Ni, Cu) martensite transformations can be observed (Figure 1d,e). In this scenario, the Cu atoms are discharged from the dendritic gap, which results in an increase in the glass forming ability (GFA) of the residual liquid. Furthermore, the martensite interface loses coherence and undergoes amorphization (Figure 1c,d). Finally, liquid atoms that are located in the edge zone arrange into an amorphous configuration directly before nucleation and diffusion (Figure 1d). It can be observed that the Ni content in the inner B2-Ti (Ni, Cu) dendrite is significantly greater than that of the surface. Since Ni can effectively reduce the martensite phase transformation temperature in the BMGC, the inner B2 dendrites are stable (Figure 1f). Thus, the functionally graded composites with a continuously graded microstructure can be obtained when there is a temperature gradient in the material during solidification. In this condition, the amorphous surface may exhibit high strength and hardness, excellent wear resistance, corrosion resistance and biological compatibility, while the austenitic center will possess good plasticity.

### 3.2. Macro-Mechanical Properties

From the XRD pattern of the post-fracture sample displayed in Figure 1a, the stress-induced preferentially-oriented-(−111) martensitic transformation from the (012) partial austenite forms during loading. Figure 2a shows the work hardening rate, *θ* = d*σ*/d*ε*, and the true stress versus true strain curves of the alloy. The relationship between the work hardening rate and the true strain can be divided into three stages. The initial work hardening stage, shows that the work hardening rate sharply decreases with the increase of strain. This stage can be regarded as the preparation condition for the work hardening, and the size of this stage indicates the degree of difficulty in which the material work hardening is. Furthermore, in this stage, the alloy undergoes a phase transition in which dislocations are formed that promote the work hardening in the alloy. The second stage consists of the main stage for the strengthening and toughening of the material, where the work hardening rate decreases with the increase of the strain. The longer the second stage is, the larger the work hardening rate will be. The third stage refers to the linear hardening stage, which is characterized by a work hardening rate that monotonically decreases with the increase of strain. Compared to the other stages, the change of the work hardening rate during this stage is the slowest. In this stage, the strengthening process will end, and the longer the stage is, the larger the work hardening rate will be. That is to say, the higher the strength of material as well as the better plasticity. Figure 2b gives the true compressive stress versus strain data of Ti-based dendrite-composite alloys [15,16], and some other advanced metastable alloys [17,18,19]. It is remarkable that the DSMC BMGCs deal better with the strength–ductility trade-off.

### 3.3. Micro-Mechanical Properties

To better understand the connection between the macro-mechanical behavior and the microstructure of the alloy, the nanofretting behavior and creep strain in different regions of the sample were examined. Additionally, the energy dissipation factor (EDF) was calculated. Nanofretting is the special friction mode in which the relative reciprocating motion amplitude is in the nanometer scale. The area of a hysteresis loop in the load–displacement curve of radial nanofretting before closure reflects the process of the energy dissipation.

The maximum elastic strain energy during loading is defined as the total deformation energy (*W_L_*), which is the area under the load–displacement curve from Figure 3a (1):(1)$$W_{L}=\int_{0}^{h^{\prime }}P_{L}\left(h\right)dh\;$$
where $P_{L}\left(h\right)$ and *h′* are the loading curve function and the maximum indention depth, respectively. Then the elastic recovery energy during unloading, $W_{U}$, can be calculated by the area of enclosed zone of unloading curve from Figure 3a (2):(2)$$W_{U}=\int_{h_{f}}^{h^{\prime }}P_{U}\left(h\right)dh$$
where $P_{U}\left(h\right)$ is the unloading curve function. Therefore, the energy dissipated in each reaction is [20]:(3)$$EDF=\frac{\int_{0}^{h^{\prime }}P_{L}\left(h\right)dh-\int_{h_{f}}^{h^{\prime }}P_{U}\left(h\right)dh}{\int_{0}^{h^{\prime }}P_{L}\left(h\right)dh}\;$$

According to the load–displacement (L–D) curve results and the Oliver–Pharr method, the calculated nano-hardness in the edge, transition and central regions increases, whereas the plastic deformation, energy consumption factor and creep show the opposite trend.

### 3.4. Corrosion and Friction Behavior

Figure 4a gives the potentiodynamic polarization curves for the ASW and PBS of the Ti–6Al–4V (TC4) and Ti_40_Ni_40_Cu_20_ BMGC. Both alloys show a spontaneous passivation behavior. The free corrosion potential *E_corr_* values of the TC4 and BMGC were −0.490 and −0.346 V in the ASW at 298 K. In the PBS, these values were found to be −0.3608 and −0.287 at 310 K. As compared with the TC4, the BMGCs exhibit higher *E_corr_* values, such that it is not easy to lose electrons, demonstrating a lower tendency to corrode. The self-corrosion current density (*I_corr_*) is given by the Tafel extrapolation method, where the higher polarization resistance (*R_p_*) can be calculated using the Stern–Geary equation [21]:(4)$$R_{P}=\frac{\beta_{a}\beta_{c}}{2.3I_{corr}\;\left(\beta_{a}+\beta_{c}\right)}\;$$
where *β_a_* is anodic Tafel slope and *β_c_* is cathodic Tafel slope. *I_corr_* and the *R_p_* of the TC4 and the BMGC are 2.53 μA·cm^−2^, 1.5 × 10^5^ Ω·cm^2^ and 1.601 μA·cm^−2^, 1.9 × 10^6^ Ω·cm^2^ in the ASW, while they were 0.488 μA·cm^−2^, 4.6 × 10^5^ Ω·cm^2^ and 0.343 μA·cm^−2^, 3.2 × 10^6^ Ω·cm^2^ in the PBS. Lower *I_corr_* and higher *R_p_* values indicate that the alloy has lower thermodynamic tendency as well as slower kinetic rates of corrosion.

The standard electrode potential for the Ti, Ni and Cu elements are −1.63, −0.257 and −0.342 V, respectively [22]. Thus, it is easier for the Ti atom to lose an outer electron which results in the rapid formation of the TiO_2_ oxide film, the n-type semiconductor, hindering the environmental and body hazardous element Ni from resolving-out [23]. The oxide film is generated via the following chemical process:Ti + 2H_2_O = TiO_2_ + 4H^+^ + 4e (5)
TiO_2_ + 2H_2_O = Ti^4+^ + 4OH^−^(6)

When the generation–resolution rate is equal, the reactions reach equilibrium and a stable oxide film is formed. A higher *R_p_* value indicates the formation of a more compact passivation film with higher resistance. As the homogeneous amorphous surface seldom contains defects, non-uniform corrosion does not occur easily in the BMGC, and the corrosion-resistance is greater than that of the crystalline alloy [24].

However, the TiO_2_ film is fragile and can easily be locally eroded if defects exist in the microstructure below the passive film. Defects such as grain boundaries, inclusions, second phase precipitates, micro pores and cracks will provide a condition for the initiation of pitting corrosion in the medium with activated anions. In Figure 4a, the potentiodynamic polarization curve for the BMGC in the ASW fluctuates around 0.2 V. Also, in Figure 4b, the friction coefficient curve in the ASW also shows obvious fluctuations. In the scratched oxide film, defects are absorbed by the anions, leading to anodic dissolution on the bare alloy surface. In the protective area of the oxide film, the cathode reaction occurs, and can be written as [25].
2H_2_O + O_2_ + 4e = 4OH^−^ (pH ≥ 7)(7)

Thus, the active-passive cell forms, and the Ti ions in located in the etching pits continually increase in number. Furthermore, more Cl ions migrate into the pits. Consequently, the TiCl_4_ phase forms as the corrosion product that subsequently causes hydrolyzation. The hydrolyzation reaction is defined as:TiCl_4_ + (x + 2)H_2_O = TiO_2_·xH_2_O + 4HCl(8)

The pH value of the etching pits decreases as the reaction proceeds, leading to accelerated pitting. As illustrated in Figure 4a, the corrosion current density increases after pitting occurs in the artificial seawater. Meanwhile, the concentration of the newly formed TiO_2_ increases to a certain extent, which hinders further development of the pitting. Thus, the polarization curve presents repassivation after the pitting occurs. In addition, the corrosion pit and its surrounding corrosion structure can be observed in Figure 5. As can be seen, no obvious corrosion products are found in the rest area. EDS mappings show that the O elements distribute on the surface, concentrated in the corrosion area. The Ti content is greater and with a more uniform distribution on the surface. The Cl is mainly concentrated on the boundary of the uncorroded region and the defects-caused corroded region. The Ni and Cu elements were observed to uniformly distribute outside the corrosion area.

That is, the uncorroded region is covered with a uniform oxide film. In the corroded region and its surroundings, more O and Cl elements are enriched due to the newly formed TiO_2_ and migrating Cl^−^. Ni and Cu have not participated in the corrosion process owing to the protection of the passive film.

From Figure 4b, the friction coefficients *μ* of the ASW and PBS are only 0.261, 0.183 and 0.160 in air, and show a decreasing trend. The main wear behavior of the dry friction are fatigue and abrasive damage. As for the artificial seawater, the wear mechanisms are attributed to the fatigue and pitting wear. With respect to the PBS, the same mechanism is associated with the fatigue wear. No crystallization was found to occur on the friction surface of each sample, indicating that the BMGCs exhibits good friction-reduction and anti-wear properties.

## 4. Conclusions

In the present work, Ti-based BMGCs were designed via TRIP effect of DSMCs to enhance both the ductility and work-hardening capability. Based on the macro- and micro-mechanical behavior, as well as three stages of work hardening, the TRIP effect depends on the consistency of diffusion and moving velocity of solute atoms and movable dislocations. Moreover, the excellent wear and corrosion resistance were attributed to the amorphous surface. The versatile DSMC BMGCs can serve as advanced metals with excellent mechanical and functional properties. However, the development and application of DSMC BMGCs are limited by size restrictions and their processing and forming abilities, which need further research in the future.

## Figures and Tables

**Figure 1 materials-15-07304-f001:**
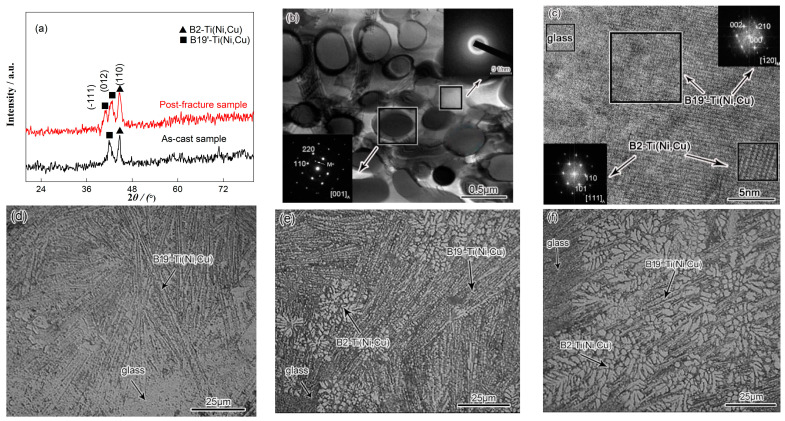
Microstructure of *Φ*3 mm Ti_40_Ni_40_Cu_20_ alloy. (**a**) XRD patterns of as-cast and post-fracture samples. (**b**) TEM and SADP of as-cast samples, dispersion halo ring and diffraction patterns confirm the amorphous and DSMC mixed structure. (**c**) High resolution transmission electron microscopy (HRTEM) and Fast Fourier transform (FFT) further certified the microstructure of DSMC BMGCs. (**d**–**f**) Optical microscope (OM) of the edge, transition and inner zones; (**d**) contains a more amorphous featureless phase, fine lath martensite and some small austenite plates; (**e**) contains more coarse lath martensite and some fine austenite dendrites, and a small amount of amorphous phase; (**f**) mostly contains the developed austenite dendrites, and little lath martensite and amorphous featureless phase.

**Figure 2 materials-15-07304-f002:**
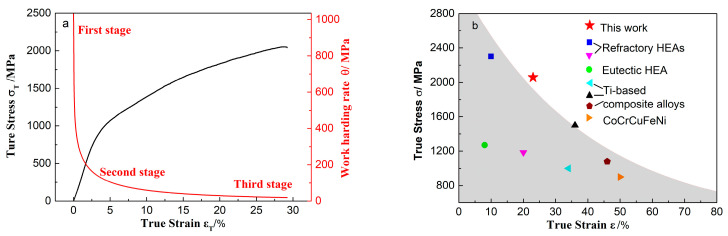
(**a**) The work hardening rate and true stress versus true strain curves engineering stress-strain curve of *Φ*3 mm Ti_40_Ni_40_Cu_20_ alloy; (**b**) compressive properties compared to some other advanced alloys.

**Figure 3 materials-15-07304-f003:**
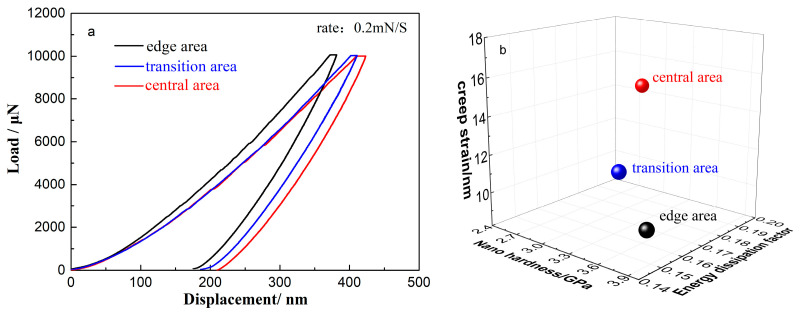
(**a**) P-H curves of nanofretting; (**b**) nano hardness, energy dissipation factor and creep strain in different areas.

**Figure 4 materials-15-07304-f004:**
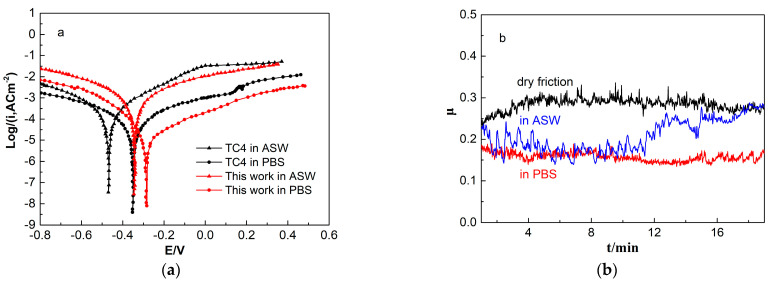
(**a**) Potentiodynamic polarization curves of Ti–6Al–4V and Ti_40_Ni_40_Cu_20_ BMGC in ASW at 298 K and PBS at 310 K; (**b**) friction coefficient of Ti_40_Ni_40_Cu_20_/GCr15 under dry friction and in ASW, PBS solutions.

**Figure 5 materials-15-07304-f005:**
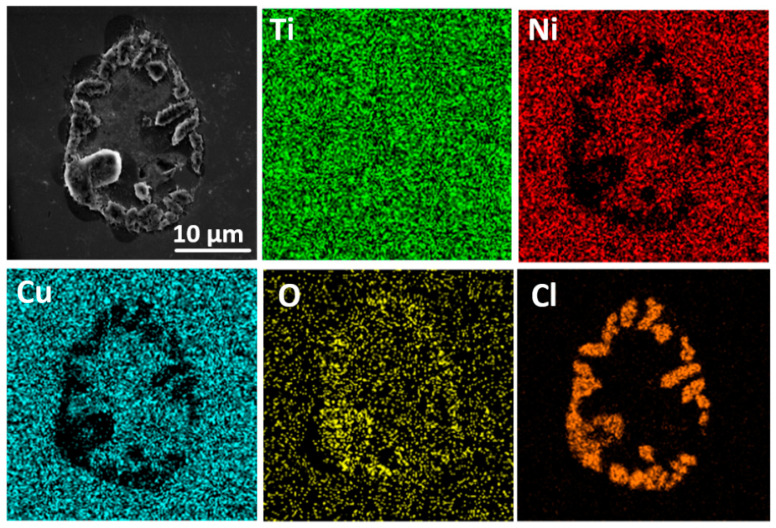
SEM and EDS mappings of Ti_40_Ni_40_Cu_20_ BMGC wear surface in ASW.

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
