# Peer review of "Versatile Medium Entropy Ti-Based Bulk Metallic Glass Composites"

_materials, 2022, doi:10.3390/ma15207304_

Round 1

Reviewer 1 Report

Please see the enclosed file.

Reviewer 2 Report

·   Discuss more about practical applications compared with the literature.

·       Add more information in introduction about Ti-based alloys.

·       For a complete characterization and corelate with the XRD analysis, it’s recommended to add microstructural analyses.

·       In the Introduction section, the authors cited the specific results of previous research and cited them adequately. However, they did not mention their shortcomings in previous research. In the Introduction section, the penultimate paragraph should contain common features of previous research. The shortcomings of previous research should also be pointed out, in general.

·       Very few recent references on dental implants: You can add also [1] Mechanical Characterization and In Vitro Assay of Biocompatible Titanium Alloys; [2] In-depth assessment of new Ti-based biocompatible materials

·       In the Introduction section, the last paragraph should contain the scientific contribution and scientific hypotheses of your research. Complete, further elaborate the scientific contribution and scientific hypotheses of your research. Be explicit. In addition to the goal of the research (which was written), the novelty in the context of the scientific contribution should be pointed out. Scientific contributions should be written based on the shortcomings of previous research in the literature. In this way, the authors will better emphasize novelty and scientific soundness.

·       In the conclusions, state the scientific contribution, the shortcomings of your methodology and future research.

Check the english. 

Round 2

Reviewer 2 Report

Paper was improved.